# Active Learning via Classifier Impact and Greedy Selection for Interactive Image Retrieval

## Abstract

Active Learning (AL) is a user-interactive approach aimed at reducing annotation costs by selecting the most crucial examples to label. Although AL has been extensively studied for image classification tasks, the specific scenario of interactive image retrieval has received relatively little attention. This scenario presents unique characteristics, including an open-set and class-imbalanced binary classification, starting with very few labeled samples. To address this specific scenario, we introduce a novel batch-mode Active Learning framework named GAL (Greedy Active Learning) that incorporates a new acquisition function for sample selection that measures the impact of each unlabeled sample on the classifier. We further embed this strategy in a greedy selection approach. We evaluate our framework with both linear (SVM) and non-linear (Gaussian Process, MLP) classifiers. For SVM and MLP, our method considers a pseudo-label strategy for each sample while ensuring tractability through a greedy approach. Considering our Gaussian Process acquisition function, we show a theoretical guarantee for the greedy approximation. Finally, we assess our performance on the interactive content-based image retrieval task and demonstrate its superiority over existing approaches and common baselines.

## 1 Introduction

Annotated datasets are in high demand for the majority of machine learning applications today. Active Learning (AL) aims to actively select the most valuable samples for annotation, that when labeled and added to the training process, will maximally boost the performance in the target task (*e.g.* a classifier). In recent years, task specific AL has gained popularity, *e.g.* for multi-class image classification (37; 13), few shot learning (3; 38) pose estimation (19), person re-identification (30), object detection (51) and interactive Content-Based Image Retrieval (CBIR) (32; 4; 34; 20).

CBIR methods play an important role for data mining in large image datasets. AL has been engaged in interactive CBIR (IIR) to reach the desired retrieval with just few user interactions. In AL cycles, the user is given a collection of unlabeled images from a pool, indicating whether each image in the collection is *relevant* (positive, belongs to the query concept) or *irrelevant* (negative). These images are then added to the training set, with their corresponding (true) label to train a new classifier for retrieval. The idea is to learn/recognize the user intent through an *iterative* and *interactive* process, as it is difficult to specify queries directly and explicitly. In the context of Content-Based Image Retrieval (CBIR), this task involves a form of active learning known as "pool-based" active learning (31), where the learner has access to a pool of unlabeled data and can request the user's label for a certain number of instances from within that pool. For image retrieval, the unlabeled pool would typically comprise the entire searched database or a subset of it. In general, and for the image retrieval task in particular, there are two main requirements for the learner in this context. Firstly, the learner must accurately grasp the target concepts. Secondly, it must achieve a quick understanding of a concept with just a small number of labeled instances. This is because the active learning process typically commences with only one or a few query examples provided by the user, and it should yield satisfactory results within a few rounds of labeling.

Selecting a batch (set) of images at each cycle (iteration) is referred to as Batch Mode AL (BMAL) (30; 27; 49; 13; 52; 37). This approach differs from the extraction of a *single* sample batch (27; 45). A general pipeline describing the process of AL for image retrieval with user feedback (IIR) is shown in Fig. 1. In each cycle, a binary classification task is introduced, characterized by highly imbalanced classes and an open-set scenario (where the categories in the search domain are typically unknown). The negative class in general consists of irrelevant images from diverse and heterogeneous classes (asymmetric scenario). Active learning methods are typically employed for classification problems where the training set is substantial, and the classes are evenly distributed. However, in the context of CBIR, the classes are highly imbalanced, with the relevant class being significantly smaller (around 20 to 100 times) than the irrelevant class. Additionally, CBIR tasks often involve open-set scenarios and asymmetric classes. As a consequence of these factors, the boundary of a binary classifier that separates the desired query concept from other samples becomes very inaccurate, especially during the initial iterations of relevance feedback. This is primarily due to the significantly reduced size of the training set in the early stages of the active learning process. In this context, numerous methods tend to be inefficient, resulting in a somewhat random selection process.

Selection strategies in active learning (AL) typically aim to predict the "value" of a specific unlabeled instance based on various hypotheses, especially in terms of increasing classification accuracy or the retrieval measure. Commonly used cues for this purpose include *uncertainty* (12; 17; 47) and *diversity* (6; 42). Hybrid approaches integrate both concepts for improved performance (48; 53; 1; 37). Nevertheless, uncertainty relies on the accuracy of the classifier, which, in turn, necessitates a sufficient number of labeled samples. These labeled samples are often unavailable, especially in the initial cycles of IIR. On the other hand, diversity is solely based on the distribution of samples, but it may not always lead to the selection of the most effective points. In response to these challenges, our method proposes a new approach that incorporates both uncertainty and diversity cues in a more effective manner.

Active learning (AL) models have been extensively studied across various tasks and conditions. For instance, they have been explored in the context of *cold start*, where the initial labeled training set is small (21; 54; 55; 18). Additionally, AL has been applied to address issues such as class imbalance, rare classes, and redundancy, as demonstrated in SIMILAR (29). The BADGE model, as introduced in (2), effectively balances diversity and uncertainty without the need for any hand-tuned hyper-parameters, much like our approach. Additionally, certain methods specifically target large batch sizes, aiming to reduce the number of training runs required to update heavy Deep Neural Networks (DNNs). For instance, ClusterMargin (10) addresses the presence of redundant examples within a batch.

Recent studies such as (21; 54) have investigated the influence of budget size on active learning strategies (diversity vs. uncertainty) and have also addressed the challenge of cold start for balanced multi-class classification tasks. In the context of cold start, poor results are attributed to the inaccuracy of trained classifiers in capturing uncertainty, a problem that becomes more pronounced with small labeled training sets (35; 16). Active learning methods designed for cold start in image classification typically begin the process with a few tens (for small datasets like CIFAR10) to hundreds of samples per class (for datasets like CIFAR-100 and ImageNet-100) (21). Moreover, they often operate in a closed-set scenario with equal sized classes, where type of categories are known beforehand, and samples from all classes are provided at the start. This scenario differs from IIR tasks, which involve binary classification and an open-set context, where the number and types of classes in the search space are unknown.

**Motivation:** In the context of a cold start scenario, where the labeled dataset is exceptionally limited, the active learning procedure becomes notably more challenging. The complexity stems from our inability to rely on the classifier to estimate the label or uncertainty of a candidate data point. This situation is a common practice in active learning for AL-CBIR. Moreover, there is the open-set classification challenge, that involves dealing with unknown classes. The proposed GAL algorithm addresses these challenges by two aspects: 1) A greedy method that best exploits the few labeled samples available and gradually enlarges the train set within the batch cycle. 2) Using an acquisition function that prioritizes data points that have the most significant impact on *reshaping the decision boundary*, or the *global uncertainty measure* (in contrast to sample uncertainty). We therefore change the common hypothesis that relies on parameters estimated from a (weak) classifier (e.g., uncertainty or direct prediction), to an approach that focuses on the *impact of individual samples on the classifier*. This approach is better adapted to AL for a binary classifier with few

positives and unknown open-set negatives. We demonstrate the behavior of our method through clear toy examples with scarce labeled samples showing the tendency of our method in fusing between uncertainty and diversity. Remarkably, this approach better manages the scarcity of labeled samples and the diversity of categories within the dataset, since it considers the *change* in the classifier.

To this end, we propose a batch mode active learning method for interactive image retrieval (IIR) that inherently incorporates a cold start, in an open-set scenario. Traditional AL methods designed for standard image classification can become impractical under such circumstances, due to model instability and unreliable uncertainty estimation (55; 25; 36). We hereby focus on each individual sample and propose three types of acquisition functions for AL sample selection, that measure the global change in the boundary decision. For a linear/non-linear classifier (SVM/MLP), we measure the impact on the decision hyper-plane if the label is flipped. When considering a non-linear Gaussian Process classifier, we evaluate the impact of each sample on the change in the overall uncertainty of the classifier (in contrast to direct sample uncertainty in previous works). To further cope with the scarcity of labeled samples, we suggest a *greedy* scheme that efficiently exploits each sample in the subsequent selection of each batch. Our approach effectively combines both uncertainty and diversity, as demonstrated in Section 3. Our work can be related to the MaxMin approach (27); however, we extend and generalize this idea by introducing a flexible framework that can be adapted to different classifiers and accommodate a larger budget size. This is achieved through a novel acquisition function within the proposed greedy method.

## 2 Related Work

Pool-based AL for Image Information Retrieval (IIR) can be defined as a binary (or one-class) classification task with several unique characteristics: (i) Open-set: The number of classes and their categories in the pool are unknown. (ii) Imbalance: Often, less than 1% of the pool contains the query concept (positive class). (iii) Cold start: Only a few labeled samples are available, particularly in the early and crucial cycles. However, many existing AL methods are not specifically designed and tested in scenarios that combine several characteristics, such as *cold start*, imbalance, rare classes, and an open-set scenario. In the context of pool-based IIR, several initial studies have proposed the use of a tuned SVM with either engineered or deep features (44; 20; 34; 40). SVM offers a practical approach for dealing with small training sets, as it possesses a strong regularizer. For instance, in (44), a kernel SVM classifier is utilized for binary classification tasks.

Gosselin *et al.* (20) proposed RETIN, a method that incorporates boundary correction to improve the representation of the database ranking objective in CBIR. In (34), the authors introduced an SVM-based batch mode active learning approach that breaks down the problem into two stages. First, an SVM is trained to filter the images in the database. Then, a ranking function is computed to select the most informative samples, considering both the scores of the SVM function and the similarity metric between the "ideal query" and the images in the database. A more recent work by (40) addresses the challenges related to the insufficiency of the training set and limited feedback information in each relevance feedback (RF) iteration. They begin with an initial SVM classifier for image retrieval and propose a feature subspace partition based on a pseudo-labeling strategy.

Zhang *et al.* (56) proposed a method based on multiple instance learning and Fisher information, where they consider the most ambiguous picture as the most valuable one and utilize pseudo-labeling. In contrast, Mehra *et al.* (32) adopt a semi-supervised approach, using the unlabeled data in the pool for classifier training. They employ an uncertainty sampling strategy that selects the label of the point nearest to the decision boundary of the classifier, which is based on a heuristic of adaptive thresholding. To enhance their results, they incorporate semantic information extracted from WordNet, requiring additional textual input from the user. On the other hand, Barz *et al.* (5) proposes a method called **ITAL** that aims to maximize the mutual information (MI) between the expected user feedback and the relevance model. They utilize a non-linear Gaussian process as the classifier for retrieval. Kapoor *et al.* (26) introduced an AL technique employing Gaussian processes for object categorization. In each cycle, the method selects a single point—specifically, an unlabeled point characterized by the highest uncertainty in classification. This uncertainty is assessed by taking into account both the minimum posterior mean (closest to the boundary) and the maximum posterior variance. Zhao *et al.* (57) introduced an efficient Bayesian active learning method for Gaussian

Process Classification. In this procedure, one sample is chosen in each cycle unlike our methodology, which aims to minimize overall uncertainty. Moreover, our method doesn't rely on knowledge about the distribution of the negative set, which can be highly multimodal due to the presence of various class types.

In this work, we introduce a novel AL algorithm for Image Interactive Retrieval (IIR). We follow the common strategy in the existing few shot learning methods *e.g.* (43; 9) and IIR (34; 5) by learning deep features from a large, labeled dataset (such as ImageNet), and then employing a "shallow classifier" (in terms of adjustable parameters) to avoid overfitting at cold start. We first, introduce two new acquisition functions focusing on the *change* in the boundary decision or global uncertainty, instead of a sample uncertainty. We illustrate the advantage of our acquisition function in fusing uncertainty and diversity concerns. Next, instead of relying on a batch-based selection strategy for the top-n most influential samples, which may underperform when dealing with small training sets, we adopt a *greedy* approach. To achieve this goal for our SVM and MLP classifier, we add selected points, with their pseudo labels to the training set, as the batch stack is accumulated. In the case of Gaussian Process, the algorithm seeks the next sample that minimizes the overall uncertainty, a strategy that solely depends on the samples, eliminating label-related issues.

To summarize, we present an innovative approach to Batch Mode Active Learning (BMAL) for IIR tasks with the following contributions:

1. We propose new acquisition functions that quantify the *impact value* on the classifier as a selection strategy, tailored to both linear and non-linear classifiers. Our framework is adaptable to different classifiers, where, for instance, the impact value can measure the global shift in the decision boundary or the level of global uncertainty of the classifier.

2. We propose a novel greedy scheme, to cope with very few labeled samples, focusing on only one class, and operates in an open-set regime with highly imbalanced classes.

3. For the Gaussian Process-based classifier, we demonstrate a lower bound on the performance of the greedy algorithm, using the $(1 - 1/e)$-Approximation Theorem.

4. We present a more realistic multi-label benchmark for the Content-Based Image Retrieval (CBIR) task, named FSOD, where the query concept involves an object within the input image.

5. We evaluate our framework using three classification methods (linear and non-linear) on four diverse datasets, showcasing superior results compared to previous methods and strong baselines.

## 3 Algorithm Description

This section presents our GAL (Greedy Active Learning) framework, which employs a greedy approach for active learning. We showcase its application in three scenarios: linear and two non-linear classifiers. The fundamental concept behind the greedy approach is to select the optimal sample $x^*$ (and its corresponding pseudo-label $\hat{l}^*$ in the SVM and MLP cases) from a candidate set, based on maximizing a score $\mathcal{S}$ which we referred to as an *impact value*. Subsequently, $x^*$ and $\hat{l}^*$ are added to the labeled set, and the process repeats to select the next sample. This iterative procedure persists until a designated budget $B$ is reached. The greedy approach ensures that similar samples are not redundantly included in the budget set. Once a sample $x_i$ is added, the algorithm proceeds to find the next optimal point $x_j$, given that $x_i$ is already in the training set.

We follow the common strategy in few-shot learning where features are a-priori learned on a large labeled corpus (*e.g.* ImageNet). We then follow the assumption where all the images in the dataset are represented by feature vectors $x_i \in \mathbb{R}^d$, (where $d$ is the feature dimension) either engineered or coming from a pretrained network. In this paper we derive our image features from a pre-trained backbone. Let $\mathcal{X}_u := (x_1, x_2, \ldots, x_m)$ denote the set of *unlabeled* image features (representing the searched dataset), and $\mathcal{X}_l := (x_{m+1}, x_{m+2}, \ldots, x_{m+l})$ the *labeled* set. *Relevant* (positive) and *irrelevant* (negative) samples are labeled by $y_i \in \{+1, -1\}$ respectively, and the label set is denoted by $\mathcal{Y}_l$. The initial labeled set $\mathcal{X}_l$ which defines the query concept, consists of few (usually 1-3) query image features labeled by $+1$. In the course

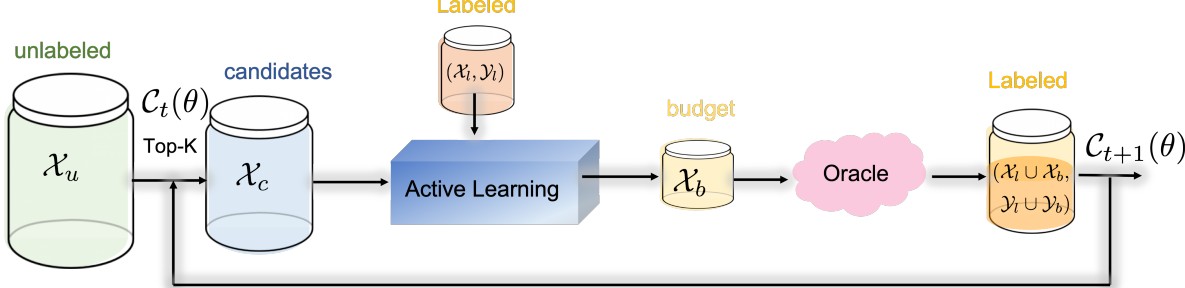

Figure 1: Main flow of the AL cycle. The top-K candidate set at cycle $t$ determined by the classifier $\mathcal{C}_t(\theta)$, can be selected as the pool from the unlabeled/search corpus. The AL module extracts a batch set $\mathcal{X}_b$ which is sent for annotation by a user (oracle) that generates the label set $\mathcal{Y}_b$. Based on the extended training set, a new classifier $\mathcal{C}_{t+1}(\theta)$ is trained for the next cycle.

of the iterative process, the user receives an unlabeled batch set $\mathcal{X}_b \subset \mathcal{X}_u$ of size $B := |\mathcal{X}_b|$, and is asked to label the relevant ($y = +1$) and irrelevant ($y = -1$) images. The AL procedure selects the set of $B$ samples, such that when labeled and added to the training set, aims to reach the maximum retrieval performance. In this work, we suggest a *greedy-based framework* which consists of two phases at each AL cycle. Let $\mathcal{C}_t$ be the classifier at cycle $t$. In the first phase, a candidate subset $\mathcal{X}_c \subseteq \mathcal{X}_u$ of size $K := |\mathcal{X}_c|$ is selected out of the unlabeled pool. This set can be either the whole unlabeled dataset or a subset which is determined by the top-K relevance probabilities. The candidate set $\mathcal{X}_c$ accommodates mostly irrelevant samples due to the natural data imbalance. In the second phase, the algorithm extracts a batch set $\mathcal{X}_b \subset \mathcal{X}_c$ by an AL procedure. A user (oracle) annotates the images selected in $\mathcal{X}_b$ and adds their features and labels into the labeled set $(\mathcal{X}_l, \mathcal{Y}_l)$. Based on the new training set, a classifier $\mathcal{C}_{t+1}$ is trained for the next cycle, as illustrated in Fig. 1.

The selection process is designed to pick the samples which are mostly "effective" upon being labeled, *i.e.* maximally improve the classifier performance. At each greedy step, an *impact value* of each unlabeled sample is computed, evaluating the contribution of the sample to the classifier improvement, and the sample with the highest impact value is added to $\mathcal{X}_b$ as described in Algorithm 1. We now demonstrate the GAL framework in three settings: linear (SVM) and non-linear (Gaussian Process and MLP) classifiers via the greedy approach.

---

**Algorithm 1** Greedy Active Learning (GAL) Algorithm

---

    **function** GAL($\mathcal{X}_c, \mathcal{X}_l, \mathcal{Y}_l, B$)
        $\mathcal{X}_b \leftarrow \{\}$
        **for** $i \leftarrow 1$ to $B$ **do**
            $x^*, \hat{l}^* \leftarrow$ NEXT($\mathcal{X}_c, \mathcal{X}_l, \mathcal{Y}_l$)                ▷ Find the point that maximizes the impact value $\mathcal{S}$
            $\mathcal{X}_l \leftarrow \mathcal{X}_l \cup \{x^*\}$
            $\mathcal{Y}_l \leftarrow \mathcal{Y}_l \cup \{\hat{l}^*\}$
            $\mathcal{X}_c \leftarrow \mathcal{X}_c \setminus x^*$
            $\mathcal{X}_b \leftarrow \mathcal{X}_b \cup \{x^*\}$
        **end for**
        **return** $\mathcal{X}_b$
    **end function**

---

### 3.1 Linear Classifier

Let us define the outcome of a trained *binary* classifier $C$ parameterized by $\theta$, as the measure for the relevance of a sample to a query image. Effective or prominent samples are those that apply the most influence on the classifier's decision boundary. These sample points play a significant role in the active

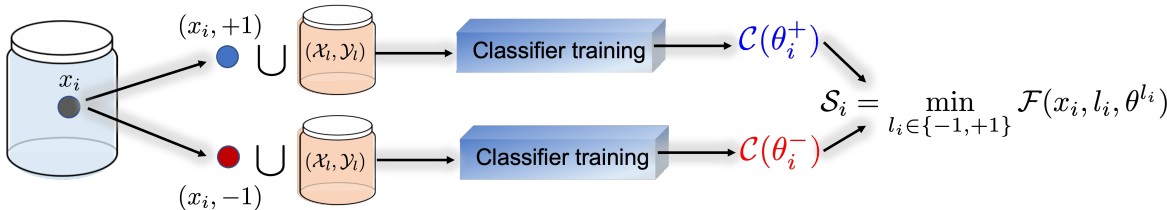

Figure 2: To calculate the score for a point $x_i$ in the candidate set, we train a classifier $\mathcal{C}(\theta_i^+)$ by assuming the sample is positive. Similarly, we train another classifier $\mathcal{C}(\theta_i^-)$ with a negative label. The impact value $\mathcal{S}_i$ is then determined as the minimum value obtained by applying a function $\mathcal{F}$ to both options (4).

learning process, shaping the classifier's evolution across iterative cycles. However, two primary challenges emerge with this approach: (i) When dealing with a search space that may encompass millions or even more samples, computational efficiency becomes a critical concern. (ii) Due to the scarcity of labels, a shallow classifier such as SVM linear classifier is favored. Additionally, SVM has a strong regularizer to avoid an overfit. Such a classifier also enables relatively rapid training durations. It's important to mention that a single-layer feed-forward neural network (NN) can also be utilized, as it is equivalent to Logistic Regression and is expected to produce outcomes similar to those of SVM. However, the use of a multi-layer perceptron (MLP) in our specific context carries the risk of overfitting due to the limited size of the training dataset, potentially resulting in increased computational overhead during the search procedure.

Additionally, we restrict our examination to samples within the candidate set, denoted as $x \in \mathcal{X}_c$, which is notably smaller than the entire dataset. Regarding the second issue, given the absence of true labels, we employ *pseudo labels*. The core principle of our proposed algorithm is rooted in the MaxMin paradigm, where we aim to MAXimize the MINimal shift in the decision boundary. This minimal shift serves as an approximation for the true label and is thus treated as a pseudo label. The underline assumption is that when an incorrect label is introduced into the training data, it may have a more substantial impact on the classifier's decision boundary, causing it to shift significantly.

Let us assume that $x_i$ has a label $l_i$, and $\theta^{l_i}$ represents the parameters of a classifier as if the point $x_i$ is included in the training set with label $l_i$. One possible impact value could be the quantification of the decision boundary's change when $x_i$ is added to the training set. Let $W_0 \in \mathbb{R}^d$ define the initial SVM hyperplane of the AL cycle, and $W \in \mathbb{R}^d$ the hyperplane which was obtained with an additional candidate point $x_i$ with label $l_i$. We then define an acquisition function as

$$\mathcal{F}_{svm} := \|W(x_i, l_i) - W_0\|_2^2. \tag{1}$$

Note that theoretically, there are two unknowns involved in this process. The label, and the most effective point $x^*$ given the label. Ideally, if the labels of the candidate points were known, then

$$x^* = \underset{x_i \in \mathcal{X}_c}{\operatorname{argmax}} \mathcal{F}_{svm}(x_i, l_i, \theta^{l_i}), \tag{2}$$

and $l^*$ is the label of the optimal point. This selection is conditioned on the sample label which is unavailable in practice. We therefore suggest to estimate the label by the minimizer of $\mathcal{F}_{svm}$ such that

$$\hat{l}_i := \underset{l_i \in \{-1,+1\}}{\operatorname{argmin}} \mathcal{F}_{svm}(x_i, l_i, \theta^{l_i}). \tag{3}$$

We refer to $\hat{l}_i$ as a *pseudo-label*. The impact value is therefore defined as

$$\mathcal{S}_i := \mathcal{F}_{svm}(x_i, \hat{l}_i, \theta^{\hat{l}_i}) = \underset{l_i \in \{-1,1\}}{\min} \mathcal{F}_{svm}(x_i, l_i, \theta^{l_i}), \tag{4}$$

where we essentially search for samples that mostly impact the decision boundary as a proxy for the most effective sample. The underline assumption relies on the separability of the data. We hypothesize that a false label will lead to a larger change in the decision boundary. This is not desirable for the selection, since

the importance of the point might be spurious. The true label though, leads to a smoother and moderate behavior. Fig. 4a illustrates this hypothesis: let the dashed line be the current boundary (based on the train set). Representing the unlabeled candidates as transparent points, we pick a sample (green point) and assign it with the wrong label (blue), to generate a new decision boundary (blue line). Similarly, the true negative (red) label yields a classifier indicated by the red line. We observe that the incorrect label results a higher deviation from the current classifier ($\mathcal{C}_t$) as expected. The index of the selected point is then given by the largest impact value among the candidate points,

$$i^* = \underset{i \in 1,2,\ldots,|\mathcal{X}_c|}{\operatorname{argmax}} \mathcal{S}_i, \tag{5}$$

where

$$\mathcal{S}_i = \min_{l_i \in \{-1,+1\}} \mathcal{F}_{svm}(x_i, l_i, \theta^{l_i}). \tag{6}$$

This selection procedure, denoted by NEXT, is summarized in Algorithm 2 and Fig. 2.

### 3.1.1 Greedy Approach

The ultimate objective of the AL procedure is to extract a batch consisting of $B$ samples. Ideally, the optimal solution would search for all the permutations of positive and negative labels of the candidate set such that the impact value would be maximal. This is of course intractable. We therefore use the greedy active learning (GAL) approach which is illustrated in Fig. 3. In GAL, the sample $x_{i_0}$ is initially selected by NEXT (Algorithm 2). We then insert its pseudo label into the train set, and calculate the next optimal point $x_{i_1}$. In this illustration, $\hat{l}_0 = +1$ associated with the left child of the tree root. At the third iteration $\hat{l}_1 = -1$ and $i_4$ is selected. Samples $i_0, i_1, i_4$ (marked by the red circles in Fig. 3) are then inserted into the budget set $\mathcal{X}_b$. This procedure continues recursively until the budget $B$ is reached, as described in Algorithm 1.

---

**Algorithm 2** Selecting the Next Point

---

**function** $\text{NEXT}(\mathcal{X}_c, \mathcal{X}_l, \mathcal{Y}_l)$
    **for** $i \leftarrow 1$ to $|\mathcal{X}_c|$ **do**
        $x_i \leftarrow \mathcal{X}_c[i]$
        **if** SVM **then**
            $\theta^+ \leftarrow \text{Classifier}(\mathcal{X}_l \cup x_i, \mathcal{Y}_l \cup +1)$                                     ▷ SVM
            $\theta^- \leftarrow \text{Classifier}(\mathcal{X}_l \cup x_i, \mathcal{Y}_l \cup -1)$
            $\hat{l}_i \leftarrow \text{argmin}_{l_i \in \{-1,+1\}} \mathcal{F}_{svm}(x_i, l_i, \theta^{l_i})$ by (1) and (6)
            $\mathcal{S}_i \leftarrow \mathcal{F}_{svm}(x_i, \hat{l}_i, \theta^{\hat{l}_i})$
        **else if** GP **then**
            $\mathcal{S}_i \leftarrow \mathcal{F}_{gp}(x_i)$ by (12)                                          ▷ Gaussian Process
            $\hat{l}_i \leftarrow Null$
        **end if**
    **end for**
    $i^* \leftarrow \arg\max_i \mathcal{S}_i$
    **return** $x_{i^*}, \hat{l}_{i^*}$
**end function**

---

In this study we focus on the few labeled set setting (early interactive cycles), where the algorithm is highly dependent on the pseudo-label evaluation accuracy. The points are selected according to Algorithms 1,2 based on (1). We demonstrate the selection pattern of our GAL with SVM in Fig. 4b for $B = 3$. In this toy example, there is an imbalanced dataset of points in $\mathbb{R}^2$ with two relevant (solid blue) and seven irrelevant (solid red) samples. Updating the classifier (yellow line) with the added selected points (green circles) we show the shift of the current classifier (dashed line) towards the solid black line which represents the optimal classifier (trained with the whole dataset). Moreover, the proposed GAL implicitly encodes both *uncertainty* and *diversity* concepts as the selection of the two points near the boundary can be related to uncertainty and the distant point to diversity (see Fig. 4b). The uncertainty is a by product of the MaxMin operator (5), (6).

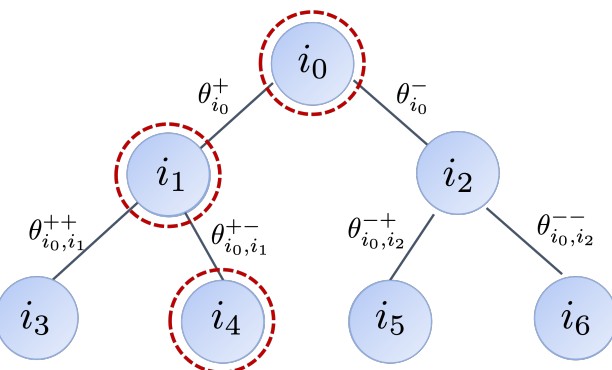

Figure 3: In the SVM scenario, the GAL algorithm employs a binary tree structure. The initial point $x_{i_0}$ is chosen through the NEXT procedure (Algorithm 2). The red circles represent the results obtained from NEXT, which are based on the corresponding pseudo-labels.

.

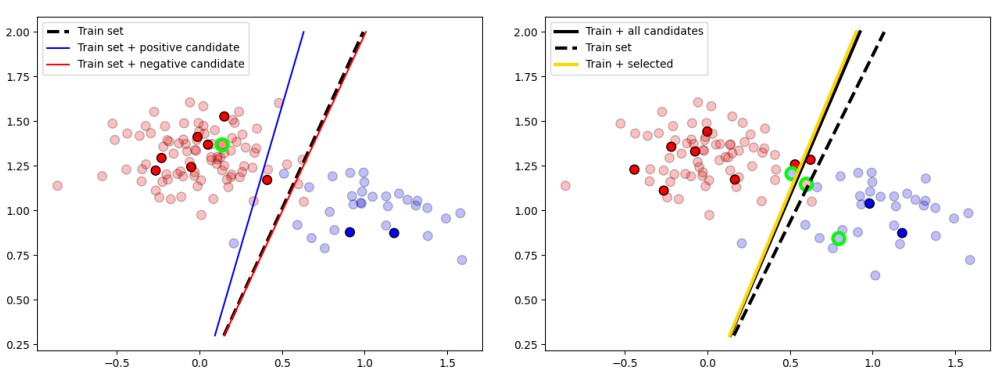

(a)                                              (b)

Figure 4: (a) Label proxy demonstration. Points sampled from two Gaussian distributions. A Change in the decision boundary for two label options are shown. Red/Blue stands for negative/positive labels respectively. Bold/light points indicate train/candidate samples with the corresponding labels. The dashed line is the classifier based only on the train set (bold circles). Blue and red lines designate the emerging classifier as if the selected point (green circle) is labeled as blue or red. The red classifier has a lower deviation from the dashed line which is in accordance with the true label (red). (b) GAL selection of three points (green circles) with two positive (blue) and one negative (red) samples. Two points are close to the classifier boundary (implying uncertainty), and the third is further away from both (implying diversity). The new classifier (yellow line) gets closer to the target boundary (bold black line) which is based on the whole dataset.

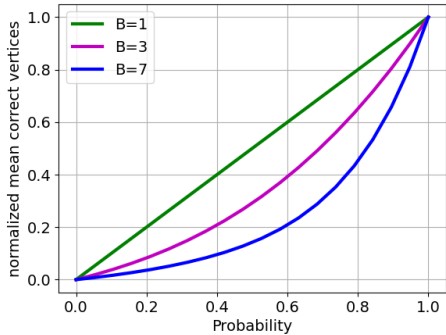

Figure 5: The normalized probability of obtaining $B$ accurate pseudo-labels vs. the probability of correctly estimating one pseudo-label.

Points with high uncertainty (close to the boundary) will likely cause the maximum change in the separating hyperplane and therefore will be selected by (1) as can be seen in Fig. 4b. As for diversity, selection of nearby samples in the embedding space are discouraged due to our approach. Note that whenever a sample point is added to the labeled set, selection of a similar point will result in a low impact value and will be dropped due to the Max operation, promoting selection of distant points (Fig. 4b).

Another theoretical aspect of the algorithm relies on the budget size $B$. The suggested algorithm is highly dependent on the pseudo label $\hat{l}$, where the effectiveness of the AL algorithm increases as the pseudo labels become more reliable. Let $p$ be the probability for correct pseudo label. The normalized probability, denoted as $P_N$, of obtaining $B$ accurate pseudo labels is given by

$$P_N = \frac{1}{B} \sum_{i=1}^{B} p^i. \tag{7}$$

The normalized probability $P_n$ (7) is plotted in Fig. 5 for different $B$ values and correct pseudo labels probabilities. It naturally suggests that a larger batch size is more sensitive to errors, while a smaller value of $B$ is preferred in each active learning (AL) cycle. This rationalization will be demonstrated in the experimental results.

### 3.1.2 Complexity for SVM-Based GAL

Lastly, the complexity of training a linear classifier such as SVM is approximately $O(dn^2)$, where $n$ is the number of samples and $d$ is the feature dimension (8). Hence, the complexity of our algorithm at cycle $i$ with $K$ candidates and a budget $B$ is given by

$$\text{Complexity}(i) = \mathcal{O}(BKd(iB)^2). \tag{8}$$

### 3.2 Nonlinear Classifier: Gaussian Process

Gaussian Processes (GP) (50) are generic supervised learning method designed to solve regression and probabilistic classification problems where the prediction interpolates the observations. Classification or regression by means of a GP, is a non-linear and non-parametric procedure that does not require iterative algorithms for updating. In addition, GP provides an estimate of the uncertainty for every test point, as illustrated in Fig. 6. As can be seen, uncertainty (pink region) is significant as we get further away from the the train (black) points. A Gaussian process can be thought of as a Gaussian distribution over functions $f : \mathcal{X} \to \mathbb{R}$, where in our case $f(x)$ represents the decision boundary. GP is fully specified by a mean function $\mu : \mathcal{X} \to \mathbb{R}$ and a covariance function $\Sigma : \mathcal{X} \times \mathcal{X} \to \mathbb{R}$ (also known as a kernel function). The mean function represents the expected value of the function at any input point, while the covariance function determines

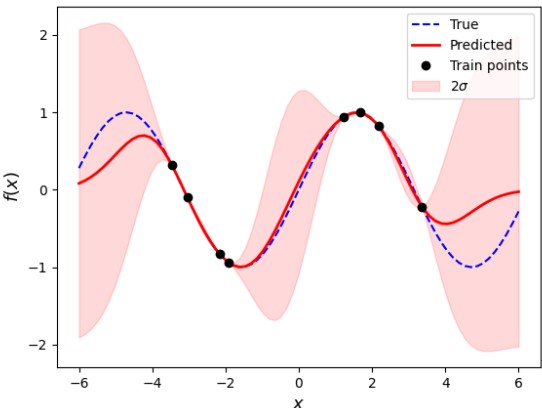

Figure 6: Gaussian Process: The true function is represented by a dashed blue line, while the prediction based on the training points is depicted by the red line. The uncertainty (std) of the prediction is illustrated by the pink area, and the training points are denoted by black circles.

the similarity between different input points. The Squared Exponential Kernel is defined as

$$\mathcal{K}(x, x') = \exp\left(-\frac{1}{2\gamma^2}\|x - x'\|^2\right). \tag{9}$$

Let $\mathcal{A} := \mathcal{X}_l$ be the train set of size $L$, and $\mathcal{X}_c$ the candidate set of size $K$. The training kernel matrix is defines as $\Sigma_{11}(\mathcal{A}) \in \mathbb{R}^{L \times L}$ where every entry in the matrix is given by (9) for $x, x' \in \mathcal{A}$. Similarly, the train-test kernel matrix is defined as $\Sigma_{12} \in \mathbb{R}^{L \times K}$, $x \in \mathcal{A}, x' \in \mathcal{X}_c$, and test kernel matrix is given by $\Sigma_{22} \in \mathbb{R}^{K \times K}$, $x, x' \in \mathcal{X}_c$. Then, the mean function is expressed by

$$\mu_{\mathcal{A}} = \Sigma_{12}^T \Sigma_{11}^{-1}(\mathcal{A}) f(\mathbf{x}), \ \mathbf{x} = [x_1, x_2, \ldots] \in \mathcal{A},$$

and the covariance matrix is given by

$$\Sigma_{\mathcal{A}} = \Sigma_{22} - \Sigma_{21}\Sigma_{11}^{-1}(\mathcal{A})\Sigma_{12}. \tag{10}$$

The variance at test point $x_i'$ is given by the diagonal term

$$\sigma_{\mathcal{A}}^2(x_i') = \Sigma_{\mathcal{A}}(i, i). \tag{11}$$

Equation (10) reflects the *variance reduction* of the test set due to the train set $\mathcal{A}$. In our setting, $\mu_{\mathcal{A}}(x_i)$ and $\sigma_{\mathcal{A}}^2(x_i)$ denote the decision boundary (red curve in Fig. 6), and uncertainty (pink area in Fig. 6) at point $x_i$ given the train set $\mathcal{A}$. In the AL procedure, our goal is to identify samples that minimize the overall uncertainty. Now, At each AL cycle, if the current train set is denoted by $\mathcal{A}$, we define the acquisition function of a candidate point $x_i$ as the uncertainty area as if $x_i$ was added into the train set,

$$\mathcal{F}_{gp}(x_i) := -\left(\sum_{x \in \mathcal{X}_c} \sigma_{\mathcal{A} \cup x_i}^2(x) + \alpha \max_{x \in \mathcal{X}_c} \sigma_{\mathcal{A} \cup x_i}^2(x)\right). \tag{12}$$

The first term describes the global extent of uncertainty across $\mathcal{X}_c$ in the integral or average sense and is therefore insensitive to abrupt changes in the pointwise variation of $\sigma^2(x)$. On the other hand, the second term represents the $L_\infty$ norm, $\|\sigma^2(x)\|_\infty$ which is designed to manage potential points of discontinuity or large deviations that we aim to minimize. Samples which maximize this function are considered informative[1]. Note that by (10), the uncertainty covariance does not depend on the labels of the training set, avoiding the problem of pseudo labeling. The NEXT algorithm for the GP is described in Algorithm 2.

---

[1]The minus sign is used to change the min to max operator.

### 3.2.1 Theoretical Analysis

We now investigate the conditions which guarantee a reasonable good approximation to the optimal batch selection. Nemhauser *et al.* (33) established a performance lower bound for a greedy algorithm when employed to maximize a set function. Let $B \in \mathbb{N}$ be a budget, $\mathcal{X}$, a finite set and a set function $F(\mathcal{A})$ with $\mathcal{A} \subseteq \mathcal{X}$. For the following maximization problem

$$\mathcal{A}^* = \underset{|\mathcal{A}| \leq B}{\operatorname{argmax}} F(\mathcal{A}),$$

the greedy algorithm returns

$$F(\mathcal{A}_{\text{greedy}}) \geq \left(1 - \frac{1}{e}\right) F(\mathcal{A}^*).$$

under the following conditions:

1. $F(\mathcal{A}) \geq 0$.

2. $F$ is non-negative and monotone, $\mathcal{A} \subset \mathcal{B}$ implies $F(\mathcal{A}) \leq F(\mathcal{B})$.

3. $F$ is submodular if for all subsets $S \subseteq T \subseteq \mathcal{X}$, and all $x \in \mathcal{X} \setminus T$, $F(S \cup x) - F(S) \geq F(T \cup x) - F(T)$.

The submodularity property has the *diminishing returns* behavior: the gain of adding in a particular element $x$ decreases or stays the same each time another element is added to the subset. By (10) and (11), the variance reduction at a test point $x_i$ is given by

$$\sigma_{\mathcal{A}}^2(x_i) := \Sigma_{22}(i,i) - \Sigma_{21}\Sigma_{11}^{-1}(\mathcal{A})\Sigma_{12}(i,i), \tag{13}$$

and the acquisition function given a train batch $\mathcal{A}$ is given by

$$F(\mathcal{A}) = -\left(\sum_{x \in \mathcal{X}_c} \sigma_{\mathcal{A}}^2(x) + \alpha \max_{x \in \mathcal{X}_c} \sigma_{\mathcal{A}}^2(x)\right). \tag{14}$$

We now show that the conditions for the $(1 - 1/e)$-Approximation theorem are satisfied for (14). The variance reduction $\Sigma_{21}\Sigma_{11}^{-1}(\mathcal{A})\Sigma_{12}$ is guaranteed to be strictly positive due to the positive-definite nature of the covariance matrix, which is an inherent property of GP modeling, and proved to be increasing monotone and submodular by Das and Kempe (11). We now use a property of submodular functions: the sum of submodular functions is also submodular (15). Based on this property, (14) is submodular as well. Employing the same considerations implies that (14) exhibits monotonic increasing behavior. Consequently, our acquisition function (14) satisfies the conditions of the $(1 - 1/e)$-Approximation theorem.

### 3.2.2 Complexity for Gaussian Process-Based GAL

Lastly, the complexity of a matrix of order $n$ inversion is $\mathcal{O}(n^3)$ and two matrix multiplications in (13) are $\mathcal{O}(n^2 K)$ and $\mathcal{O}(K^2 n)$. Hence for each AL cycle $i$ with $K$ candidates and a budget $B$,

$$\text{Complexity}(i) = \mathcal{O}\left(BK\left[(iB)^3 + K^2(iB) + K(iB)^2\right]\right). \tag{15}$$

### 3.3 Nonlinear Classifier: MLP

Let us consider a network that comprises of $L$ layers, using a non-linear activation function (ReLU). The classifier is trained using the cross-entropy loss function. As in the linear case, the impact value measures the extent of the change in the decision boundary. The AL algorithm remains identical to Algorithm 2, with the only change of replacement of $\mathcal{F}_{svm}$ with $\mathcal{F}_{mlp}$:

$$\mathcal{F}_{mlp} := \|\Psi(x_i, l_i) - \Psi_0\|, \tag{16}$$

where $\Psi$ is a vector of concatenated and flattened network weights. Specifically, $\Psi_0$ defines the initial MLP weights at the current active learning cycle, and $\Psi(x_i, i_i)$ is the weight vector as if the network was trained with $x_i$ and label $l_i$.

## 4 Evaluation

We asses the GAL framework by employing three image retrieval techniques, which utilize linear (SVM) and two non-linear (Gaussian Process, MLP) classifiers. The algorithm for the linear classifier is based on the impact value (1). In our evaluation, we compare our approach against various AL algorithms. (i) Random selection, (ii) Cyclic Output Discrepancy (COD) (22), (iii) MaxiMin (27), (iv) Ranked batch-mode AL (RBMAL) (7), and in the cases where $B > 1$, (v) Coreset (42; 28) and (vi) Kmeans++ (46). The COD (22) method estimates the sample uncertainty by measuring the difference of model outputs between two consecutive active learning cycles,

$$\mathcal{S}_{\text{cod}} := \|\mathcal{C}(x; \theta_t) - \mathcal{C}(x; \theta_{t-1})\| \tag{17}$$

where $\mathcal{C}(x)$ is the classifier prediction, $\theta_t$ and $\theta_{t-1}$ are its parameter set in the current and previous active learning cycles, respectively. MaxiMin (27) algorithm maximizes the minimum norm of the classifier, *i.e.* prioratizing "smoother" classifiers among the possible functions

$$\mathcal{S}_{\text{MaxiMin}} := \min_{l \in \{+1, -1\}} \|f(x)^l\|. \tag{18}$$

$\|f(x)^l\|$ denotes the norm of interpolating function when training the classifier with positive and negative labels of $x$. In the linear SVM case, $f(x) = \|W\|_2^2$. RBMAL method (7) combines uncertainty and diversity by

$$\mathcal{S}_{\text{RBMAL}} := \alpha(1 - \phi(x, x_{\text{labeled}})) + (1 - \alpha)u(x), \tag{19}$$

where $\phi$ is a similarity measure, $u(x)$ the uncertainty, and $\alpha = |\mathcal{X}_u|/(|\mathcal{X}_u| + |\mathcal{X}_l|)$. The batch set extracted by the above three methods, is obtained by selection of top-B score samples. Kmeans++ (46) and Coreset (28; 42) are diversity-based BMAL methods, and therefore applicable for $B > 1$. In Kmeans++, the batch samples are chosen as the closest points to each of the $B$ centroids, and in Coreset, we ensure that the batch samples adequately represent the entire candidate pool based on the $L_2$ norm distance.

In our second image retrieval approach, we incorporate a Gaussian Process (GP) technique, which was proposed in (5) and referred to as Information-Theoretic AL (ITAL). This method employs a selection strategy that aims to maximize the mutual information between the expected user feedback and the relevance model. To integrate the GP into our framework, we steer the active learning selection process towards data points that minimize the overall uncertainty of the GP classifier, as defined in equation (12).

### 4.1 Datasets

We evaluate the GAL on a wide range of scenarios including 4 datasets, representing image-level and object-level IIR. For instance-level retrieval, we used Paris-6K abbreviated as **Paris**, following the standard protocol as suggested in (39). This dataset contains 11 different monuments from Paris, plus 1M distractor images, resulting in 9994 images with $51 - 289$ samples per class and 8204 distractors. Next, we built a benchmark based on Places365 (58), indicated as **Places**. It contains 365 different types of places such as "restaurants", "basements", "swimming pools" etc. Our Places dataset consists of the validation set of Places365. We used 30 classes as queries (randomly sampled) with 100 samples per-class. Lastly, we validated ourselves on object-level retrieval, a previously unexplored task in CBIR-AL. To this end we built a new benchmark from the FSOD dataset (14), often used for few-shot object detection tasks. At this benchmark images often include multiple objects (labels), therefore introducing a high challenge for a retrieval model. FSOD dataset is split into base and novel classes. We used the base set, for our benchmark. The base set contains $5, 2350$ images with 800 objects categories where each object appears in 22-208 images. As our query pool, we randomly chose 30 object categories appearing in 50-200 images. We refer to this dataset as **FSOD-IR** and we intend to share the protocol publicly for future research. In all the above experiments, we used a Resnet-50 backbone pre-trained on Imagenet-21K (41). For the first iteration we used the top-K nearest neighbors by the cosine similarity. We used one query for Paris and Places benchmarks, and two queries for FSOD-IR (due to multiplicity of objects in images). We repeated the process for 5 random queries and calculated mAP at each AL cycle.

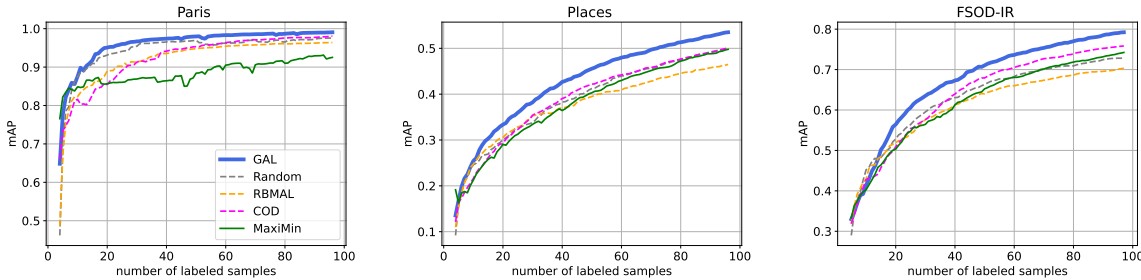

Figure 7: mAP Learning Curves of SVM-based GAL with $B = 1$ and $K = 200$ for different datasets.

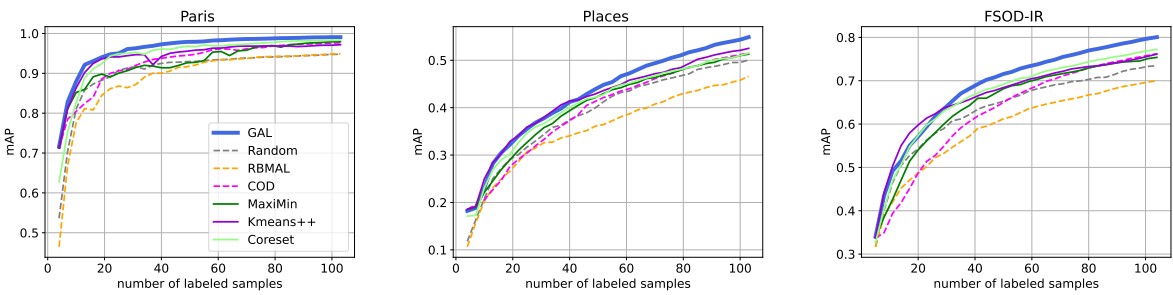

Figure 8: mAP Learning Curves of SVM-based GAL with $B = 3$ and $K = 200$ for different datasets.

To ensure a fair comparison between our method and ITAL (5) and Kapoor *et al.* (26), we conducted our evaluation of the GAL framework on the identical dataset of **MIRFLICKR-25K** (24), which was also employed in ITAL. We followed the same protocol used in ITAL for consistency. This benchmark designed for retrieval consists of 25K images, with query images belonging to multiple categories. We further used the same feature extractor as ITAL (see (5)). For all datasets we follow the same protocol: sample a query image from a certain class, consider all images belonging to that class (or containing the same object in FSOD-IR) as relevant, while instances from different classes are considered irrelevant. In all our experiments we run with 5 different initial queries for each class and report mean average precision (mAP) as retrieval performance.

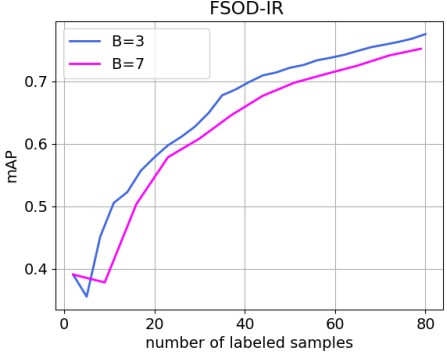

Figure 9: mAP Learning Curves of SVM-based GAL with $B = 3$ and $B = 7$. It is evident that the larger batch size yields inferior results.

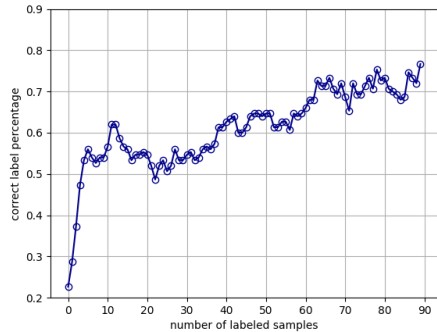

Figure 10: Pseudo-label accuracy tested on FSOD Benchmark, averaged over all classes and for candidate size of 200 and $B = 1$. Random choice is 50%.

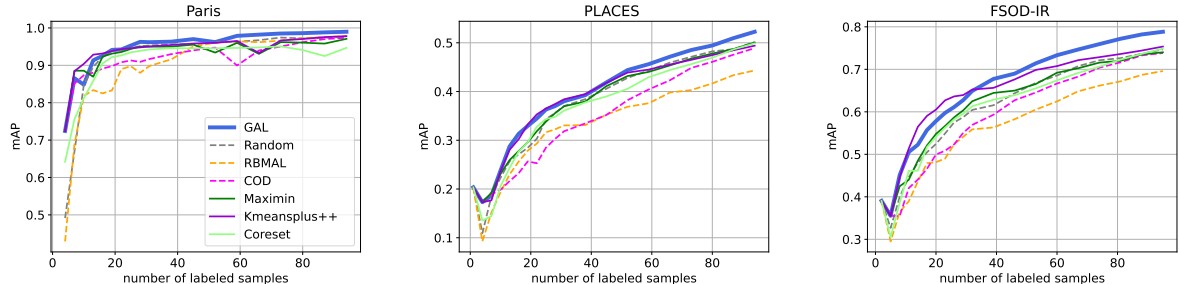

Figure 11: mAP Learning Curves of SVM-based GAL with $B = 3$ followed by $B = 7$ and $K = 200$ for different datasets.

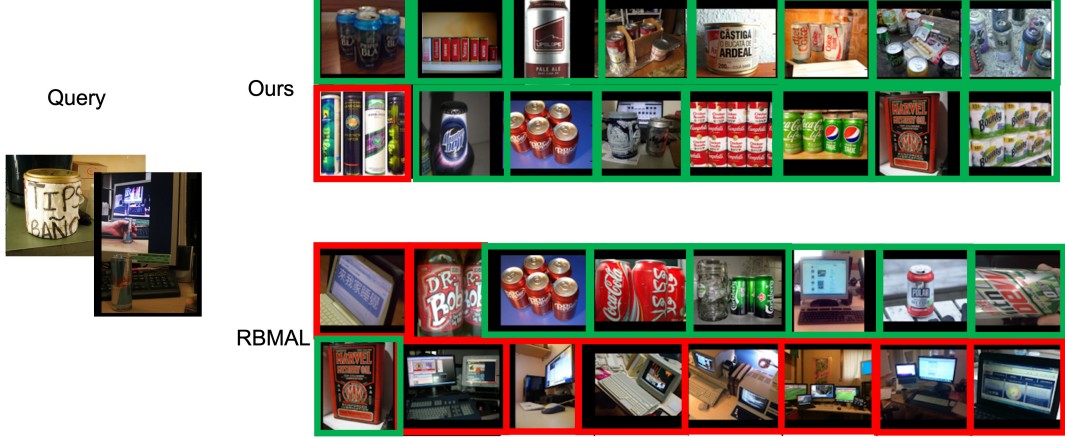

Figure 12: Image retrieval results for *Tin Can* in FSOD-IR dataset with $B = 3$ at iteration 4. Green boxes stand for relevant results while red boxes account for false positives. The second query image has two objects: Can and Display monitor. The RBMAL method mistakenly retrieves images with monitor, where GAL succeeds to find the common pattern in the queries. This example illustrates how the initial ambiguity regarding the object is gradually resolved through the active learning cycles, allowing the algorithm to effectively capture the query concept.

## 4.2 Experimental Results

We quantified the AL methods by their learning curves, indicating the retrieval performance (measured in mAP) progress along the interactive cycles. The curves are then aggregated by a single measure of the *Normalized Area under Learning Curve* (5) between 1,2 to 95 labeled samples. The results for both SVM and GP are averaged over five different randomly selected queries. We further found that the strategy of selection from a pool of top-K ranked samples according to the classifier obtained from the previous round is beneficial in our GAL and often also in the competitive methods. This subset $\mathcal{X}_c$ is comparably rich of positive samples and 'hard' negatives, further reducing the extreme imbalance in the general dataset. For instance, our experiment on FSOD showed that in average, 30% of the candidate set included positive samples, comparing to 0.5% density in the general dataset. We further analyze the impact of different candidate set size $K$ up to the whole unlabeled dataset.

| | Paris | | | | Places | | | | FSOD | | | |
|---|---|---|---|---|---|---|---|---|---|---|---|---|
| Candidate size | 100 | 200 | 1k | all | 100 | 200 | 1k | all | 100 | 200 | 1k | all |
| Random | 0.847 | 0.942 | 0.834 | 0.810 | 0.375 | 0.390 | 0.298 | 0.224 | 0.576 | 0.630 | 0.452 | 0.404 |
| RBMAL (7) | **0.915** | 0.920 | 0.806 | 0.731 | 0.410 | 0.375 | 0.293 | 0.217 | 0.660 | 0.610 | 0.466 | 0.390 |
| COD (23) | 0.909 | 0.924 | 0.881 | 0.716 | 0.399 | 0.391 | 0.359 | 0.221 | 0.630 | 0.639 | 0.606 | 0.410 |
| MaxiMin (27) | 0.883 | 0.885 | 0.892 | - | 0.395 | 0.381 | 0.363 | - | 0.625 | 0.621 | 0.603 | - |
| GAL (ours) | 0.903 | **0.960** | **0.960** | - | **0.428** | **0.426** | **0.418** | - | **0.674** | **0.672** | **0.672** | - |

Table 1: Normalized Area under Learning Curve with $B = 1$ under different candidate settings. These results indicate the influence of our impact value of the selected samples. We indicate the top performing method in bold and the second place by the underline mark. We omit the test results for "all" in several cases due to increased computation cost and saturation.

| | Paris | | | | Places | | | | FSOD | | | |
|---|---|---|---|---|---|---|---|---|---|---|---|---|
| Candidate size | 100 | 200 | 1k | all | 100 | 200 | 1k | all | 100 | 200 | 1k | all |
| Random | 0.922 | 0.905 | 0.812 | 0.807 | 0.402 | 0.388 | 0.283 | 0.217 | 0.637 | 0.633 | 0.473 | 0.404 |
| RBMAL (7) | 0.923 | 0.888 | 0.785 | 0.718 | 0.397 | 0.355 | 0.295 | 0.213 | 0.652 | 0.592 | 0.467 | 0.389 |
| COD (23) | 0.914 | 0.927 | 0.895 | 0.692 | 0.394 | 0.394 | 0.351 | 0.213 | 0.625 | 0.627 | 0.605 | 0.398 |
| Kmeans++ | 0.922 | 0.941 | 0.935 | 0.744 | 0.416 | 0.417 | 0.394 | 0.205 | 0.661 | 0.666 | 0.632 | 0.393 |
| Coreset (28) | 0.915 | 0.943 | 0.914 | 0.767 | 0.405 | 0.407 | 0.357 | 0.230 | 0.664 | 0.666 | 0.599 | 0.418 |
| MaxiMin (27) | 0.906 | 0.926 | 0.916 | 0.906 | 0.409 | 0.402 | 0.368 | - | 0.657 | 0.648 | 0.612 | - |
| GAL (ours) | **0.946** | **0.960** | 0.952 | - | **0.430** | **0.427** | **0.419** | - | **0.681** | **0.686** | **0.675** | - |
| GAL (batch) | 0.943 | 0.957 | **0.955** | - | **0.431** | 0.421 | 0.417 | - | 0.679 | 0.678 | **0.675** | - |

Table 2: Normalized Area Under Learning Curve with $B = 3$, under different candidate settings. We indicate the top performing method in bold and the second place by the underline mark. GAL(batch) shows the result of our approach without the greedy component of our scheme.

As an ablation study we conducted tests to evaluate the impact of our suggested acquisition function for AL selection and also tests on our algorithm under non-greedy settings by selecting the top-B samples that maximize the impact values (1), (6) and (12) given a budget $B$. The non-greedy approach may encounter issues with redundant samples, as similar points could have similar scores. In contrast, the greedy algorithm prevents this scenario by ensuring that once a sample is selected, it is added to the training set. This allows for the selection of a new sample that maximizes the score function, taking into account the updated training set.

|  | Paris | Places | FSOD |
|---|---|---|---|
| Random | 0.905 | 0.388 | 0.633 |
| RBMAL (7) | 0.888 | 0.355 | 0.592 |
| COD (23) | 0.927 | 0.394 | 0.627 |
| Kmeans++ | 0.941 | 0.417 | 0.666 |
| Coreset (28) | 0.943 | 0.407 | 0.666 |
| MaxiMin (27) | 0.926 | 0.402 | 0.648 |
| Entropy | 0.903 | 0.329 | 0.586 |
| GAL (ours) | **0.960** | **0.427** | **0.686** |
| GAL (batch) | 0.957 | 0.421 | 0.678 |

Table 3: Normalized Area Under Learning Curve with $B = 3$, $K = 200$. We indicate the top performing method in bold. Entropy shows a selection by the distance to the decision boundary.

|  | Paris | | | Places | | | FSOD | | |
|---|---|---|---|---|---|---|---|---|---|
| Candidate size | 100 | 200 | 500 | 100 | 200 | 500 | 100 | 200 | 500 |
| Random | 0.908 | 0.908 | 0.910 | 0.344 | 0.348 | 0.316 | 0.593 | 0.582 | 0.580 |
| RBMAL (7) | 0.906 | 0.876 | 0.811 | 0.332 | 0.310 | 0.281 | 0.590 | 0.534 | 0.487 |
| COD (23) | 0.900 | 0.909 | 0.897 | 0.332 | 0.320 | 0.318 | 0.555 | 0.559 | 0.552 |
| Kmeans++ | 0.913 | 0.935 | 0.919 | **0.374** | 0.363 | 0.357 | 0.611 | 0.622 | 0.603 |
| Coreset (28) | 0.900 | 0.902 | 0.880 | 0.347 | 0.342 | 0.326 | 0.583 | 0.581 | 0.569 |
| MaxiMin (27) | 0.910 | 0.925 | 0.919 | 0.355 | 0.353 | 0.323 | 0.589 | 0.591 | 0.563 |
| GAL (ours) | **0.929** | **0.939** | **0.932** | 0.366 | **0.369** | **0.369** | **0.618** | **0.625** | **0.612** |
| GAL (batch) | **0.930** | **0.941** | 0.927 | 0.366 | 0.361 | 0.361 | **0.619** | 0.614 | **0.615** |

Table 4: Normalized Area Under Learning Curve with $B = 3, 7$, under different candidate settings. We indicate the top performing method in bold and the second place by the underline mark.

### 4.2.1 SVM Classifier

We first present the global performance measure of *Normalized Area Under Learning Curve* for the SVM-based scenario, tested for budget size $B = 1$ and $B = 3$ in tables 1 and 2. It is worth noting that the results obtained when $B = 1$ allow us to assess the impact value independently from the greedy scheme. We indicate the top performing method in bold and the second place by an underline mark. Interestingly, random sampling often yields high performance. This is consistent to other AL studies in classification benchmarks in the literature, under cold-start conditions (21) (as a diversity based strategy). Yet, in 8 out of 9 tests, GAL outperforms other methods and baselines for $B = 1$, where for $B = 3$, GAL is consistently the top performing method. Note that the top performance for all methods is reached for $K = 100$ or 200 and there is no consistent competitor in the second place, indicating the robustness of GAL approach under different candidate pools.

Another interesting observation shows that considering a larger candidate pool (from 100 to the whole dataset) does not necessarily improve the performance. Often a smaller candidate pool is preferred as observed in all the methods compared in our datasets for $B = 3$ (cf. Table 1 bottom, due to higher concentration of positive and hard negative samples, being better candidates for AL. For the majority of competitive methods, we discovered that a candidate set size of $K = 200$ is optimal and can significantly reduce the computational cost, an important aspect in an interactive system.

Next, we present a comparison of the learning curves by retrieval mean Average Precision (mAP) in figs. 7 and 8 for $B = 1$ and $B = 3$ with $K = 200$. These figures show the superior performance of GAL over previous methods and various baselines. The strongest competitor at $B = 3$ is found to be Kmeans++ which is purely based on diversity, performing comparably to GAL in low the extreme cold start (up to 25 in

FSOD-IR and up to 40 in Places). This result is consistent with the analysis in (21) showing that diversity based models such as Kmeans++ or Coreset are top performing methods at extreme cold start. Yet, as more labels are accumulated, Kmeans++ under-performs GAL that leverages also uncertainty. Furthermore, we note a substantial disparity, with 5-10% (absolute points) higher mAP when compared to MaxiMin (dark green) and around 5% better (from *e.g.* 0.75 to 0.80 in FSOD) compared to Kmeans++.

We conducted an additional investigation using a pure uncertainty-based method, in which the selection criterion involved identifying samples that are positioned closest to the decision boundary. This was achieved by selecting points greedily based on maximum entropy, referred to as *Entropy*. The results for budget size $B = 3$ and $K = 200$ are presented in Table 3. It is evident that the results obtained using this Entropy method are considerably inferior to those of GAL across all the datasets. This experiment further strengthens our claim that GAL effectively combines both diversity and uncertainty. Methods that solely rely on one of these aspects tend to exhibit lower performance.

As illustrated in Fig. 9 and supported by our earlier analysis presented in Fig. 5, larger budget sizes present more significant challenge, especially during the initial cycles. The challenge is demonstrated in Fig. 10. During the initial cycles, the pseudo-label accuracy is inadequate, leading to accumulated errors, particularly for larger values of $B$. In response to this challenge, we conducted experiments where we set $B = 3$ for the first 10 cycles, which was increased to $B = 7$. Nevertheless, our method is superior to other approaches, as shown in Table 4 and Fig. 11. It is noteworthy that overall although Kmeans++ performed better in the first 10 cycles, our methods still showcase superior performance overall. The greedy approach has a slight impact in the linear SVM case, assumably due to unreliable pseudo-labels, mostly occurring at the initial cycles (see fig. 10). This strategy is better manifested in the GP process, that is label independent.

Next, we present a qualitative result displayed in Figure 12. We take two query images belonging to the 'Tin Can' class in the FSOD-IR dataset and showcase the top-16 relevant images retrieved by the GAL and RBMAL methods at the fourth iteration, with a budget of $B = 3$. In the visualization, green and red boxes are used to indicate relevant and irrelevant results, respectively. It's worth noting that the right query image contains not only a 'Tin Can' but also a monitor display. GAL successfully retrieves 15 out of 16 relevant images, with one visually reasonable error. In contrast, the RBMAL method selects a few monitor images, which are exclusively present in the second query image. This example demonstrates a common challenge in CBIR when dealing with images that contain multiple objects. While there may be initial ambiguity in the query, as the active learning cycles progress and the user tags positive examples, our model excels at selecting samples that capture the user intention concept (as shared pattern between the queries) more rapidly.

Finally, despite GAL evaluating a classifier for each selection candidate, the computational cost of our method remains reasonable for several reasons.

1. We demonstrate that a small candidate set, comprising only 0.1-1% of the dataset (obtained from the classifier's top-k), is sufficient as the active learning selection pool. In many cases, this approach even yields improved performance, as evidenced in Tables 1 and 2. Consequently, there is no need to run our algorithm on the entire unlabeled set.

2. This allows for quick training and AL cycles, a practical requirement in an interactive system such as IIR.

3. Runtime performance of GAL for Paris for $B = 7$ is depicted in Fig. 13. We show results for high $B = 7$ value in order to better visualize the quadratic behaviour of the complexity. The experimental data was fitted to a second-degree polynomial (with respect to $i$) to match the complexity equation (8). Notably, the experimental results align well with the complexity equation. We further show that at reasonable iteration cycles (typically up to 10) the iteration cycle take few seconds up to several tens of seconds at the final stages (in case that the subject was not found). These measures stand for CPU without any parallelization.

4. To further expedite runtime, our training process can be distributed in a parallel manner, assigning each candidate to a separate process.

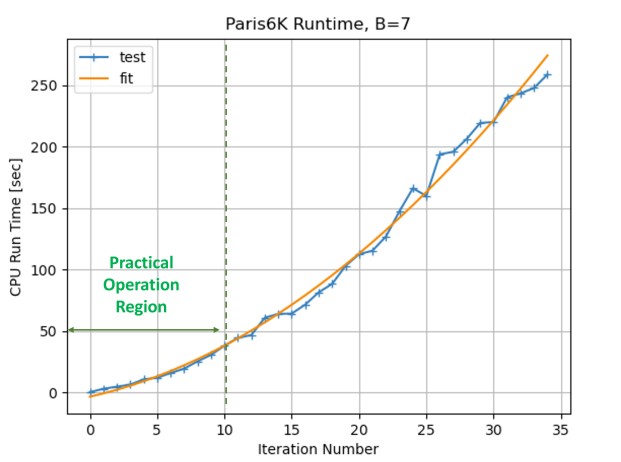

Figure 13: GAL running time [sec] for SVM on Paris benchmark, $K = 200$, $B = 7$. We show results for high $B = 7$ value in order to better visualize the quadratic behaviour of the complexity. We also extend the range of iteration numbers to show the fit with the theoretical complexity. In practice, the search process often ends after few rounds.

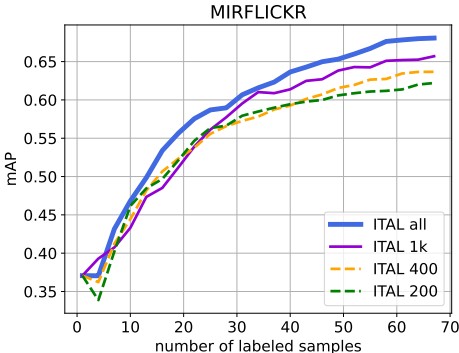

Figure 14: mAP Learning Curves of ITAL for $B = 3$ and different candidate set size $K$.

### 4.2.2 AL with Gaussian Process

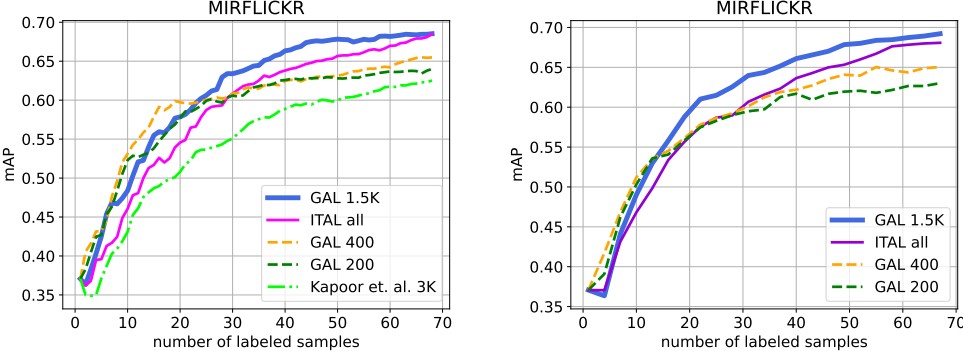

Figure 15: mAP Learning Curves of GP-based GAL with $B = 1$ (left) and $B = 3$ (right) for MIRFLICKR database. ITAL used the whole unlabeled set, while GAL and Kapoor *et al.* (26) used different candidate set size K (see Table 6).

We further present the results of GAL utilizing a Gaussian Process (GP) classifier, which are compared to ITAL (5). For this purpose, we replaced the active learning (AL) module of ITAL with GAL, employing our acquisition function (12). To make a fair comparison, we first ran ITAL with varying candidate pool sizes $K$. Fig. 14 illustrates the results of ITAL for $B = 3$ and $K = 200, 400, 1000$, as well as the entire dataset ($K = 20,000$). Table 5 provides a summary of these findings. It is evident that the entire unlabeled dataset is needed for ITAL to reach it's best result.

| $K$ | Normalized AUC |
|---|---|
| 200 | 0.547 |
| 400 | 0.552 |
| 1,000 | 0.564 |
| 20,000 | **0.585** |

Table 5: Normalized Areas under Curve of ITAL (5) algorithm for $B = 3$ at variety of candidate set sizes $K$. ITAL requires all the corpus for maximum performance.

Next, we compared GAL and ITAL. Normalized Areas under Curve are summarized in the top panel of Table 6, where GAL outperforms ITAL even when considering only 1,500 points which are 7.5% of the unlabeled dataset as candidates. We further observe the impact of our greedy scheme component boosting the overall performance by nearly 7% (from 0.566 to 0.605) with respect to standard batch selection strategy (denoted by GAL(batch), *i.e.* choosing the top-B scores at each round). Fig. 15 depicts the comparison between these two methods for $B = 1$ and $B = 3$ respectively with candidate pool K=200,400, and 1,500. The figure shows 2-5% mAP improvement with K=1,500. Running time of GAL for $B = 3$ and $K = 200$ is shown in Fig. 16. The experimental data was fitted to a third-degree polynomial (with respect to i) which is in accordance with the complexity equation (15).

Finally, we conducted a comparison between GAL and another uncertainty-based approach proposed by Kapoor et al.(26) which was designed for $B = 1$. This method aims to identify the sample which is closest to the decision boundary with the highest uncertainty $\sigma$. We adapted this approach to our framework, evaluating its performance across various values of $K$, with the optimal performance observed at $K = 3,000$. GAL consistently outperformed this method across all tested values of $K$. The summarized results can be found in Table 6 and depicted in the left part of Fig. 15.

| method | $K$ | $B = 1$ | $B = 3$ |
|---|---|---|---|
| ITAL (5) | 20,000 (all) | 0.586 | 0.585 |
| Kappor *et al.* (26) | 1,500 | 0.517 | |
| Kapoor *et al.* (26) | 3,000 | 0.542 | |
| Kapoor *et al.* (26) | 20,000 (all) | 0.457 | |
| GAL (ours) | 200 | 0.584 | 0.570 |
| GAL (ours) | 400 | 0.593 | 0.583 |
| GAL (ours) | 1,500 | **0.608** | **0.605** |
| GAL (batch) | 200 | 0.584 | 0.553 |
| GAL (batch) | 400 | 0.593 | 0.573 |
| GAL (batch) | 1,500 | **0.608** | 0.566 |

Table 6: Normalized Area under Learning Curves for MIRFLICKR database. Our GAL outperforms ITAL (5) and Kapoor *et al.* (26). Note that for $B = 1$ there is no greedy process. The impact of our greedy scheme is manifested in $B = 3$.

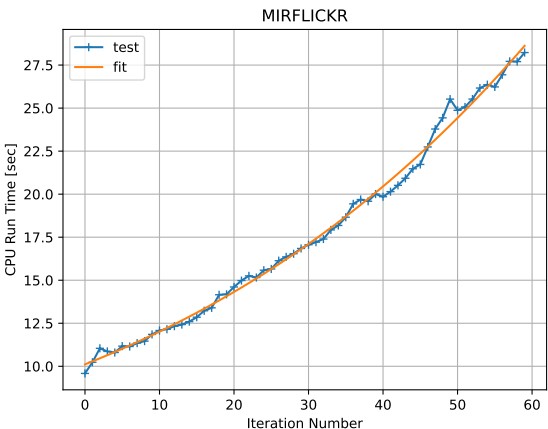

Figure 16: GAL running time [sec] for GP, $K = 200$, $B = 3$. We show the fit with the theoretical complexity (15).

### 4.2.3  AL with MLP Classifier

In this section, we present the outcomes of active learning when applied to an additional non-linear classifier. It's important to note that the classifier in the context of AL-CBIR comprises two distinct stages: (i) the sample selection strategy (AL) and (ii) retrieval. As discussed in section 3.1, it is crucial to recognize that the utilization of non-linear classifiers in retrieval tasks may lead to immediate overfitting issues, primarily due to the significantly limited size of the training dataset. We therefore extended our work by employing a three-layer MLP (10 neurons at the inner layers) with a ReLU activation function for the AL selection, while continuing to utilize the Gaussian Process (GP) method for retrieval. To make a fair comparison we used the same retrieval method of GP in all compared methods. In this setting as well, the GAL method outperformed competitive algorithms as can be seen in Fig. 17 for the MIRFLICKR dataset with $B = 3$ and $K = 200$.

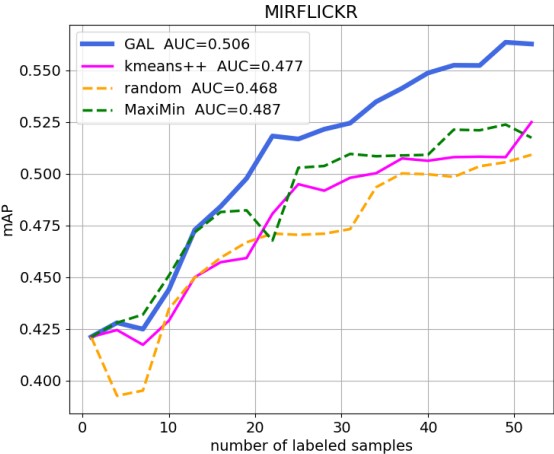

Figure 17: mAP Learning Curves of MLP-based AL selection with $B = 3$ and $K = 200$ applied on MIR-FLICKR.

## 5    Summary and Future Work

In this paper we address the problem of active learning for Interactive Image Retrieval task. This task introduces several unique challenges, process starting with only few labeled samples in hand and challenging open-set and asymmetric scenario (the negative set includes various unknown categories with different size). To cope with these circumstances we suggested a new approach that addresses the above challenges by two main aspects. First, by considering the impact of each individual sample on the decision boundary as a cue for sample selection in the AL process. To this end, our acquisition function, considers pseudo labels or directly optimizes a global uncertainty measure. Second, to better cope with the scarcity of labeled samples in a batch mode AL, we embed our approach in a greedy framework where each selected sample in the batch is added to the train set, before selecting the subsequent best promising one. This process is continued until the designated budget is reached, attempting to effectively extend the train set, within each batch. Additionally, we provide a theoretical analysis that supports the idea that our greedy scheme offers a reliable approximation in the context of Gaussian Process.

Our method was demonstrated on linear and non-linear yet efficient classifiers on several large image retrieval benchmarks, including a new challenging one including small objects. Promising results were obtained compared to previous methods. In addition, we believe that our framework can pave the way for broader applications, including the cold start problem in realistic open-set scenarios. In future work, it would be valuable to incorporate a quantitative measure that quantifies the disentanglement of the two implicitly combined cues of diversity and uncertainty. This could lead to improvements in performance and provide better explainability of the algorithm's decisions.

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
