# OpenReview forum: "Active Learning via Classifier Impact and Greedy Selection for Interactive Image Retrieval"
_TMLR — Rejected by TMLR_

### Review · Reviewer_mLAD · 2023-08-27

**Summary Of Contributions:**

This paper studies the active learning approach in interactive image retrieval task. The authors proposed a new framework, namely Greedy Active Learning (GAL), to select most informative training instances from the unlabeled instances. Two instantiations of GAL are given: linear SVM and Gaussian Process classifier. Empirical results show the proposed method achieves better performance compared with several other methods in image retrieval benchmarks,

**Audience:**

Yes

**Broader Impact Concerns:**

I have no concerns about the ethical implications of this work.

**Claims And Evidence:**

Yes

**Requested Changes:**

I think the paper should be revised to provide more justifications for the method designing.

The Theorem 3.1 seems to be known knowledge and I don't see the necessity to include that in the paper.

Please avoid using the general term 'score' when discussing some specific values. For example, I'd suggest to use a self explanatory name like "sample gain utility value" or "instance impact value" than the pain term "score" in Section 3 to avoid confusion.

**Strengths And Weaknesses:**

The paper reads ok and gives sufficient introduction on the problem setting as well as the discussion on related works. The proposed method achieves better performance than the baseline methods on several image retrieval datasets. The proposed method uses a pseudo labeled instance to evaluate the impact on the existing classifier and therefore greedily choosing the most informative instance. The greedy selection with submodular function is straightforward and not novel. However, I don't think the authors provide enough justification behind the pseudo label. Eq(3) select the pseudo label that is the current model prediction, and as a result the algorithm will choose whatever instances that is close to the decision boundary. This is because the pseudo-label will not "disagree" with the existing model. In real scenarios, an instance that is far from decision boundary and does not agree with the current model should provide the most "surprise" to the current model and add value to the training set. I wonder how is the GAL compared to naively choosing instances that is closest to the decision boundary.

---

> ### Author Response · Authors · 2023-08-30
> **Response to Reviewer mLAD**
>
> We sincerely appreciate the time and effort the reviewer dedicated to our manuscript.
> We have carefully considered your feedback and would like to express our gratitude for providing your perspective on our research.
>
> 1. **Pseudo-labels:** Please note that pseudo-labels are used only as part of the loss in our SVM classifier, and not in the Gaussian Process (as we mention in section 3.2).
> 2. **Pseudo-label justification:** The underlying concept of the proposed algorithm relies on the MAXMIN paradigm, i.e. trying to MAXimize the MINimal shift of the decision boundary. The Minimal shift provides an estimation for the true label (and is therefore a pseudo label).  These points are not necessarily around the decision boundary. We illustrate the behavior of this approach in Fig. 4. Particularly in Fig. 4b, we show that among the 3 selected points, 2 are close to the boundary, and the 3rd point is far away.  This is explained in the first part  of page 8. Our approach therefore integrates both uncertainty and diversity concepts in a particular way. The combination of uncertainty and diversity has been shown to be a key component for previous AL methods (See Hybrid approaches in the introduction [1,36,47,52] in page 2).  These ideas will be further clarified in the revised version. As for comparison to a naive approach that chooses instances that are closest to the decision boundary, we will conduct this test and provide the results of this test ASAP.
> 3. **... submodular function is straightforward and not novel:** Thank you for pointing this out. The idea of Theorem 3.1 was to introduce the background formulation and at the same time to show that our suggested cost function satisfies the conditions of this theorem (submodular set function). Note that our new loss is composed of two components L2 and L_inf. We show that the submodular terms are satisfied for this type of  loss. The core message is that our proposed model is theoretically justified.  We will rephrase this part in the revised version, by removing the Theorem, focusing on the Theorem conditions and relation to our loss for making it clear.
> 4. We will rephrase the term “cost” according to the reviewer’s suggestion.

---

### Review · Reviewer_4w3T · 2023-09-11

**Summary Of Contributions:**

This paper delves into an exploration of batch-mode pool-based Active Learning (AL), with a specific emphasis on its application within the domain of Content-Based Image Retrieval (CBIR). CBIR exhibits distinctive characteristics, prominently featuring an open-set classification environment, a cold-start scenario, and an inherent class imbalance. In this paper, the authors introduce a novel batch-mode active learning framework denoted as Greedy Active Learning (GAL). In particular, the study includes a comprehensive evaluation of GAL’s performance when employed in conjunction with both linear and nonlinear classifiers. Furthermore, GAL is subjected to testing scenarios involving class imbalance, open-set and cold start settings. Experiments indicates that GAL outperforms existing active learning approaches and common baselines on benchmark datasets of interactive content-based image retrieval tasks.

**Audience:**

Yes

**Claims And Evidence:**

Yes

**Requested Changes:**

1.	It is recommended that the authors provide a more comprehensive and detailed explanation regarding why their designed algorithm exhibits improved adaptability to open-set and cold start scenarios.

2.	It is advisable to employ a more suitable example to illustrate the point being made based on the findings presented in Fig. 12. An improved example should showcase a scenario in which the object corresponding to the label and the interfering object exhibit a significant disparity in the composition of the image.

3.	To show effectiveness on other non-linear model, it is suggested to add experiments on other non-linear models to show effectiveness of proposed method. If the method demonstrates effectiveness exclusively within the context of Gaussian Process models, the designation of “greedy active learning” for the proposed novel framework may not be entirely suitable.

4.	It would be beneficial for the authors to provide an explanation for noticeable and abrupt decline in their results shown in Fig. 11.

**Strengths And Weaknesses:**

Strength

1. One of the strength of this paper lies in the authors’ approach to active learning algorithm design, which encompasses both linear and non-linear models. They have specifically designed acquisition functions to suit the unique characteristics and requirements of each model, thereby enhancing the adaptability and effectiveness of the active learning framework.

2. Another strength of this paper is the design of ablation study. To rigorously assess the effectiveness of the proposed greedy methods, the authors conducted experiments under non-greedy settings.

3. The final commendable aspect of this paper is its rigorous theoretical analysis of the proposed method. The paper offers a theoretical guarantee for the greedy approximation, enhancing the overall credibility and robustness of the research findings.

Weakness

1. In the abstract and introduction sections, the authors indicate that their method’s application in addressing the specific scenarios of open-set classification, class imbalance, and cold start. In Algorithm 1, the candidate set X_C is specifically designed for accommodates mostly irrelevant samples due to the natural data imbalance. However, it appears that the details of how open-set and cold start scenarios are specifically taken into consideration are somewhat lacking in the subsequent description of the proposed method. Besides, the process of calculating X_C from X_U is not described clearly. This omission creates a gap in understanding and may raise questions about the foundation and validity of the approach under the target settings.

2. The analysis of normalized probability with batch size is insufficient. In Fig. 5, the conclusion drawn is indeed limited to the observation that a larger batch size results in a lower normalized probability under the same probability setting. It is important to acknowledge that this conclusion does not directly imply that a larger batch size will necessarily lead to increased sensitivity to errors.

3. The example utilized in Fig. 12 may not be ideal. In the right-bottom image within the query section, the labeling of the image as a “can” is inherently ambiguous due to the display taking up a significant portion of the image. Consequently, the selection of relevant examples with such inherent ambiguity is itself open to question.

4. The application scope of the proposed algorithm appears to be limited to Gaussian Process models. Given the existence of various non-linear models, it remains unclear how effective the proposed algorithm would be when applied to these alternative models.

5. The algorithm complexity of the method introduced in this paper appears to be quite high. Consequently, the test results for the scenario where “all samples are labeled” in ‘GAL’ exhibit challenges, resulting in missing data at the corresponding positions in Table 1.

Questions

1. In Equation (7), the algorithm’s complexity is denoted as O(i^2B^3), where i represents the number of iterations, and B denotes the batch size. However, the observed linear relationship in Fig. 13 between time consumption and the increments of both iteration i and batch size B contradicts the theoretical analysis presented in Equation (7). Why is such a discrepancy observed?

2. In the last two figures of Fig. 11, a noticeable and abrupt decline in the curves of all algorithms occurs within the range where the number of labeled samples is between 0 and 10. Could the authors explain the reason for such a sudden change?

---

### Review · Reviewer_5jBL · 2023-10-30

**Summary Of Contributions:**

This paper focuses on the active learning problem on the domain of interactive image retrieval that the algorithm selects which data instances are informative for the future training and asks supervisors for the annotation of them. The proposed algorithm greedily selects the optimal samples from the candidate set according to the classifier-specific score function measuring the uncertainty, diversity and performance. This paper instantiates the proposed framework for SVM and GP algorithms, and evaluates the frameworks on four image dataset (3 for SVM and 1 for GP), showing the best performance among active learning algorithms.

**Audience:**

Yes

**Broader Impact Concerns:**

No concern about the ethical implications.

**Claims And Evidence:**

Yes

**Requested Changes:**

[Major Comments]

* Reconsider the structure of introduction and the related work. Summary of this paper and its contribution is mixed in the related work.
* Reconsider the contribution of this paper. For example, new objective functions for a selection strategy is introduced as the first contribution, but the later section explains the framework with greedy selection as the main proposal and objective functions are supportive components for this framework.
* Reconsider the experiments, expanding active learning with GP and complexity comparison (i.e. running time).
* Some arguments in the introduction and related works are not supported in the later sections. (weakness 2)

[Minor Comments]

* Section 3.1 focuses on the quantification of the change of the decision boundary. Could you briefly introduce the other candidates for the score function if the proposal is one of possible score functions as argued?
* Could you explain the insight behind the second term of equation 11 (maximal value of the uncertainty of GP)?
* Could you restructure the proposed algorithms? Algorithm 2 and 3 share the same function of the greedy selection with different score functions.
* (Typo) In the last paragraph of the related work (last paragraph in page 3), the sentence “Rather than selecting the top-n most influential samples … Batch Mode Active Learning in IIR” is incomplete.
* (Typo) Equation 12 misses the parenthesis “)”.

**Strengths And Weaknesses:**

**Strength**

This paper proposed the greedy algorithm for AL that seems straightforward but shows outstanding performance on the experiments. Additionally, this work provides theoretical proof about the performance guarantee of the greedy active learning with GP if minimizing variance score.

**Weakness**

* Contributions and the later section do not fit perfectly. For example, this paper picks new objective functions for a selection strategy as one of contributions, but the experiment studies how the combination of greedy selection algorithm and new objective functions works against the existing works.
* The paper argues that the image information retrieval problem has characteristics of open-set, imbalance classes and cold start, but many existing AL methods are not specifically designed for these characteristics. It is not clear how the proposed algorithm is different from these AL methods with respect to the unique properties of IIR problem.
* Several methods have been proposed for active learning with GP, but there is only one baseline for the active learning experiment with GP. Please refer to other papers (i.e. [1] A. Kapoor et. al. “Active Learning with Gaussian Processes for Object Categorization” ICCV 2007. [2] G. Zhao et. al. “Efficient Active Learning for Gaussian Process Classification by Error Reduction” NeurIPS 2021.) for more supporting results.

---

> ### Author Response · Authors · 2023-11-13
> **Response to Reviewer 5jBL**
>
> We highly appreciate the positive feedback “outstanding performance on the experiments” and “additionally… provides theoretical proof about the performance guarantee”.  We further thank the reviewer for the constructive comments.
>
> **Contributions and the later section do not fit perfectly:**
> We have revised the paper to clearly include our intuition and motivation. Please see Sec. 1, Page 2.
>
> **It is not clear how the proposed algorithm is designed better for the IIR characteristics:**
> In the context of a cold start scenario, the complexity stems from our inability to rely on the classifier to estimate the label or uncertainty of a candidate data point in a cold start. Moreover, there is the open-set classification challenge, that involves dealing with unknown classes. The proposed GAL algorithm addresses these challenges by two aspects: 1) A greedy method that best exploits the few labeled samples available and gradually enlarges the train set within the batch cycle. 2) Using an acquisition function that prioritizes data points that have the most significant impact on **reshaping the decision boundary**, or the **global uncertainty measure** (in contrast to sample uncertainty). Remarkably, this approach better manages the scarcity of labeled samples and the diversity of categories within the dataset, since it considers the change in the classifier.
> To address this matter, we have implemented the following: 1) Dedicated a paragraph in the introduction and related work sections to elaborate on these points. 2) Termed our objective score as the "impact value" to accurately reflect the metric we are optimizing. Please see page 2 and 3 in the revised version.
>
> **Referring to other papers:**
> We express gratitude to the reviewer for highlighting these works, both of which have been appropriately cited in the related work section. Notably, our method is compared against 4-6 different widely used baselines and prior art for Support Vector Machines (SVM), such as COD 2021, MaxiMin 2020, and RBMAL 2017. Regarding Gaussian Processes (GP), our comparison specifically involves ITAL 2018, that is to the best of our knowledge the latest AL method with GP, which addresses our specific task of AL-CBIR.
> In reference to the work by Zhao et al. 2021, it is important to note that this paper addressed, standard multi-class classification task, and not image retrieval. Their experiments conducted on UCI dataset, use hand-crafted features and therefore significantly differ from our complex benchmarks based on deep features. Furthermore, their Bayesian classification method necessitates samples from each class to construct a likelihood measure which is impractical in **open-set** CBIR scenario. Our selection strategy indeed doesn't mandate labeled samples or knowledge of their distribution from all types of the negative categories.
> Concerning Kapoor et al. 2007, their method was originally designed for a multi-class classification problem. We adapted this approach to our framework and assessed its performance across various values of K, with optimal results observed at K = 3,000. GAL consistently outperformed this method across all tested values of K, as detailed in Table 6 and illustrated in the left part of Fig. 15. Our findings demonstrate that GAL surpasses this approach in performance as well.
>
> **Reconsider the structure of introduction and the related work:**
> We have added a unique paragraph that clarifies the motivation explained above in the revised version (page 2).
>
> **Reconsider contribution of this paper:**
> We have modified the contribution section in page 4 accordingly.
>
> **Reconsider the experiments, expanding active learning with GP and complexity comparison (i.e. running time).**
> We ran new experiments and added an additional comparison to Kapoor et al. paper. With respect to complexity analysis, we have the complexity for GAL in equation 15 and the corresponding runtime time in Figure 16.
>
> **[Minor Comments]**
> 1. We used the quantification of change in the decision boundary for our selection algorithm. Other options are traditionally, **sample** uncertainty and  diversity. Some studies use a combination of both as we elaborate in Sec. 1, page 2. Our method fuzes both uncertainty and diversity, showing superior results compared to other methods that use a different type of fusion for these two factors.
>  2. The first term describes the extent of uncertainty across X_c in the integral or average sense and is therefore insensitive to abrupt changes in the pointwise variation of $\sigma^2(x)$. On the other hand, the second term represents the $L_\infty$ norm,  which is designed to manage potential points of discontinuity or large deviations that we aim to minimize. We added this explanation in the revised manuscript.
> 3. Thank you for the comment. We merged Algorithm 2 and 3 into Algorithm 2.
> 4. We rephrased the sentence
> 5. Thank you, the typo was fixed.

---

### Author Response · Authors · 2023-11-13
**Revised Manuscript**

Please find attached the revised manuscript, wherein we have carefully incorporated all the review comments, and rearranged the paper accordingly. Notably, we have color-coded the changes in this version to facilitate easy tracking. The updated manuscript incorporates new experiments and comparisons as requested by the reviewers, along with clarifications and modifications in response to their valuable feedback. We express sincere gratitude for the constructive comments from all reviewers, which have significantly contributed to enhancing the quality of the manuscript.

---

### Decision · Action_Editor_SiW2 · 2024-01-17

**Recommendation:** Reject

**Comment:**

One issue came up in the discussion on this paper among reviewers: in the case of the linear SVM, for the first few queries, only unlabeled points within the margin could affect the decision boundary. Since you’re taking the min over all labels, points outside of the margin would have a score of 0 when assigned a pseudo label matching the current decision boundary, so they would never be chosen by the max over the points. The points that would have the largest impact on the decision boundary are those closest to the boundary (e.g., the margin size would decrease by 1/2 if a point exactly on the boundary were selected — either label +/- decreases the margin by 1/2). Therefore, wouldn’t the method choose exactly the same points as the max uncertainty heuristic, at least at first? Once there are no longer unlabeled points within the margin, then it might just switch to choosing points randomly (since all would have a score of 0?). If this is the case, then it should be clarified in the paper. If this isn’t the case, then the method should be clearly explained to avoid this confusion and explain precisely how it would select points (e.g., by visualizing the dynamic selections). It would also be good to show the scores for points selected in Fig 4b to demonstrate that the diversity point wasn’t just selected randomly (having a score equal to all others).

The reviewers also pointed out that the current complexity of the method is quite high. If it is performing sampling similar to the uncertainty heuristic, or if it can’t select unlabeled points early on that aren’t in the margin, then that would lead to a very efficient mechanism for pruning unlabeled points from consideration, and consequently reduce the complexity. It was good to include the current experiment showing that the computation is reasonable in practice, but please illustrate its runtime compared to other methods.

Because these issues require more than a minor revision, the paper is not yet ready for publication nor can it be accepted with minor revisions currently. However, because this method seems promising based on its empirical performance, I would like to STRONGLY ENCOURAGE THE AUTHORS TO REVISE THE ARTICLE AND RESUBMIT IT TO TMLR.

Minor issue:
Pg 6, 2nd to last line:  underline -> underlying

**Audience:**

Yes, the paper would be relevant to TMLR's audience.

**Claims And Evidence:**

The reviewers all agree that this paper proposed a straightforward (a good thing!) and high-performing method for active learning, supported by a variety of empirical and theoretical results. The overall recommendations from reviewers were mixed, with two reviewers advocating for weak acceptance and another advocating for rejection. Because of the disagreement, the action editor closely examined the reviews, the authors’ responses, and read the paper in detail. The novelty of the proposed method is low to moderate, but the technical contribution is reasonably solid, excluding a few issues discussed below and in the individual reviews. The authors are encouraged to correct these issues before the paper is ready for publication.

As noted by several reviewers, there are some organization issues that reduce the clarity of the paper. In particular, it isn’t clear exactly what the proposed method does at a high level until the new motivation paragraph in the introduction. Similarly, the contributions appear at the end of the related work. I suggest motivating the problem, positioning it within the related literature (as you do), then going into precisely what is novel about your approach (which is in the motivation paragraph, but isn’t really motivation), followed by the contributions (moved from related work). This needs to be more than just cutting and pasting paragraphs; they should be better integrated together. Similarly, the presentation of the approach could be cleaner. It took several reads through before I understood the method in detail. I’d suggest focusing on the high level summary of how it works, then go into the mathematical detail. Similarly, all algorithms should be accompanied by high-level explanations of how they work. The added text of the revision certainly helps, but it would be nice to have it a bit better integrated with the rest of the writing.

As pointed out by the reviewers, it isn’t clear exactly how the proposed method differs from what unlabeled points would be chosen by standard uncertainty and diversity heuristics. It is not clear how the selection in practice differs from those points that would be selected by standard uncertainty heuristics (which have substantially lower computational complexity). In addition, how this method induces diversity isn’t clearly explained or investigated empirically, which is a critical omission. There are some statements that it does induce diversity among the selection, and one example from Figure 4b that it does (without support of the scores — please see below), but it would be extremely helpful to include a more rigorous exploration, as is mentioned in the conclusion for future work. Experiments teasing apart the uncertainty and diversity would be insightful to add, as would some experiment showing the dynamic selection of points along with the boundaries to illustrate the dynamics of the method (and help differentiate it from other active learning methods).

**Resubmission Of Major Revision:**

The authors may consider submitting a major revision at a later time.